# Humidity Conditions of Aerated Concrete Walls Depending on the Type of Finishing Coating

**Mikhail Vladimirovich Frolov** [1,*] , **Valentina Ivanovna Loganina** [2] **and Elena Alekseevna Zhuravleva** [1]

1 Department of Heat and Gas Supply and Ventilation, Institute of Engineering Ecology, Penza State University of Architecture and Construction, 440028 Penza, Russia; elenazh2002@bk.ru
2 Department of Quality Management and Construction Technology, Faculty of Technology, Penza State University of Architecture and Construction, 440028 Penza, Russia; loganin@mail.ru
* Correspondence: mihail-frolovv@yandex.ru; Tel.: +7-9273772899

**Abstract:** A study was conducted to analyze how different types of plaster coatings affect the humidity levels in aerated concrete walls under varying climatic conditions. The suggestion is to utilize a specialized heat-insulating lime plaster designed specifically for finishing aerated concrete. It has been determined that the use of a specialized heat-insulating lime dry building mixture allows the temperature to be reduced at which condensate begins to fall in the enclosure by 1.5–7.0 °C. The research determined the amount of condensation that occurs during periods of negative outdoor temperatures in different climatic zones when using these walls. It has been established that the use of a specialized heat-insulating lime dry building mixture allows for some external walls to exclude the formation of condensate, while for others the amount of condensate falling out is reduced by 21.5–50.6%. Thus, it has been established that the use of a specialized heat-insulating lime dry building mixture significantly improves the moisture regime in the outer walls of aerated concrete.

**Keywords:** aerated concrete; finishing layer; humidity regime

## 1. Introduction

The extraction, transportation, and combustion of energy resources has a significant negative impact on the environment. Therefore, during the operation of buildings throughout the entire life cycle of the building, it is necessary to strive to reduce energy costs for heating. To do this, it is necessary not only to design energy-saving enclosing structures, but also to strive to ensure that excess moisture does not accumulate in the enclosing structures during operation. Excess moisture increases the thermal conductivity of building envelope materials and causes their premature destruction.

The tightening of energy efficiency requirements for newly erected buildings has caused an increase in the volume of construction using lightweight concrete, both in the construction of private houses and in the construction of apartment buildings. Currently, various types of lightweight concrete are used: air-entrained concrete, aerated concrete, and silicate concrete, etc. [1–4]. Now, in construction, aerated concrete is the most common; therefore, the humidity regime in walls made of this material is studied in detail in this work.

The construction of buildings using aerated concrete involves the application of a decorative finishing layer. To prevent excessive moisture and maintain the heat-shielding qualities of aerated concrete, it is important to consider the vapor permeability of the finishing layers [5–11]. Excess moisture in the wall not only worsens the heat-shielding qualities of aerated concrete, but also reduces its service life. Especially dangerous is the accumulation of moisture at the border of the aerated concrete's external finishing coating. If a significant amount of condensate falls in this area, the external plaster collapses, cracks, and peels off the wall.

Due to the temperature difference between the outdoor and indoor air in winter, mechanisms are launched based on the partial pressure gradient of water vapor that occurs during this time. There is a diffusion of steam through the building envelope, from the heated room to the street. To reduce the amount of condensing moisture, the materials in the wall must have less thermal conductivity and greater vapor permeability when moving from indoor air to outdoor air. Therefore, plaster intended for exterior finishing of aerated concrete buildings should be characterized by relatively high vapor permeability and low thermal conductivity. They should not interfere with the movement of moisture inside the wall.

When it comes to exterior finishing systems for cellular concrete walls, plaster compositions are used that meet the requirements outlined in regulatory documents. These compositions have a vapor permeability resistance (for thick-layer plaster-based finishes) of no more than 0.5 $m^2$ h Pa/mg, for thin-layer plaster-based finishes and finishes without plaster layers, the vapor permeability resistance should not exceed 0.2 $m^2 \cdot h \cdot Pa/mg$. Additionally, these compositions have a water absorption rate with capillary suction of no more than 0.5 kg/($m^2 \cdot h \cdot 0.5$) and an adhesion strength $R_{adg}$ (adhesion strength of the coating with an aerated concrete base) to cellular concrete of at least 0.15 MPa. The contact zone should also have a frost resistance rating of no less than F35 (the minimum number of freezing and thawing cycles, established by the standards, in the coating samples tested according to the basic methods, in which the original physical and mechanical properties are maintained within the normalized limits).

Currently, there are several plaster options available on the domestic market for finishing cellular concrete, including Ceresit CT 24 and Ceresit CT 24 Light plasters from Henkel, Knauf-Grunband, Isoputz Extra plasters from Knauf, ATLAS KB-TYNK plaster from ATLAS, "VERMIX ШН50" and "VERMIX ШН25" from REMIX, and "Pobedit perlite TM-30" from Pobedit [12–15]. Additionally, the scientific and technical literature suggests various compositions for exterior decoration of aerated concrete walls [16–19].

In our study, we have developed a dry building mixture (DBM) specifically designed for finishing aerated concrete. This composition includes fluffy lime, aluminosilicate ash microspheres, white cement, an additive based on a mixture of hydrosilicates and calcium hydroaluminosilicates, ground waste from aerated concrete production, a plasticizer, a redispersible powder, and a water repellent. The resulting finishing coating, based on this composition, exhibits a thermal conductivity coefficient (which is a coefficient, reflecting the property of a material, to conduct thermal energy) $\lambda = 0.137 \pm 0.007$ W/(m·°C) (dry), an adhesive strength $R_{adg} = 0.71 \pm 0.04$ MPa, a vapor permeability coefficient (which is a coefficient, reflecting the property of the material, to conduct water vapor) $\mu = 0.15 \pm 0.008$ mg/(m·h·Pa), and a compressive strength $R_c = 4.1 \pm 0.2$ MPa [20,21]. Also, an important advantage of the developed DBM should be attributed to the fact that its formulation uses ground waste from the production of aerated concrete. The efficient use of aerated concrete production waste will reduce the negative impact from its production on the environment.

Due to the limited knowledge on the use of a DBM in various climatic conditions, it is important to evaluate the effect of the finishing layer on the moisture content of aerated concrete. In this study, we assessed how the characteristics of the exterior finishing coating and the density of aerated concrete affect the humidity regime for the walls of buildings located in different climatic conditions: Rostov-on-Don, Voronezh, Novosibirsk, and Vorkuta. During the course of the research, the temperature at the onset of condensation ($t_{b.c.}$), the amount of condensate falling out, and the percentage increase in the moisture content in the wall structure of the aerated concrete were determined. The work investigated the humidity regime in walls made of aerated concrete D350-600. This choice of aerated concrete density is due to the fact that aerated concrete of these grades is usually used in the construction of single-layer walls made of aerated concrete.

It has been established that the use of a heat-insulating DBM contributes to a decrease in the temperature at the onset of condensation, a decrease in the amount of condensate,

and a decrease in the increase in weight moisture W, %, in aerated concrete enclosures made of grades D350-600. This contributes to the preservation of the high heat-shielding properties of the enclosing structure made of aerated concrete during its operation. It also allows you to extend the service life of the exterior finishing coatings and increase the period of non-repair operation in buildings made of aerated concrete.

The novelty of this research lies in the complex of findings obtained, reflecting the influence of the characteristics of the outer finishing coating, the density of the aerated concrete, and the climatic characteristics, on the humidity regime inside the outer walls. These findings from our study show the need for the careful selection of exterior finishes, taking into account all these factors. Also new is the method itself used for determining the amount of condensate falling out per year, taking into account the temperature at the onset of condensation $t_{b.c.}$, and the temperature during each of the months of moisture accumulation.

## 2. Materials and Methods

Now, when assessing the humidity regime of external walls, the following methods are widely used: determination of the zone of possible moisture condensation in the fence; calculation of the humidity regime based on the theory of humidity potential at the supersorption humidity of the materials; and determination of the temperature at the onset of condensation, according to the method of dimensionless characteristics, and others [22–25].

Each of the above methods has its own advantages and disadvantages. In this article, the use of a method for determining the temperature at the onset of condensation $t_{b.c}$ [26,27] is proposed. The choice of this method is due to the fact that it is possible to analyze the humidity regime in the walls of the building, focusing primarily on the design features of the fence. This will allow you to conveniently compare different designs of external walls according to the conditions in different cities.

For the internal plaster in the construction, lime–slag plaster was used with a density of 1600 kg/m$^3$, a thermal conductivity coefficient of 0.47 W/(m·°C), and a vapor permeability coefficient of 0.14 mg/(m·h·Pa). Three types of DBM were used as exterior finishes in the structure: cement–sand plaster with a density of 1800 kg/m$^3$, a thermal conductivity coefficient of 0.76 W/(m·°C), and a vapor permeability coefficient of 0.09 mg/(m·h·Pa); Knauf GRUNBAND with a density of 1100 kg/m$^3$, a thermal conductivity coefficient of 0.35 W/(m·°C), and a vapor permeability coefficient of 0.1 mg/(m·h·Pa); and a heat-insulating lime–cement DBM developed with a density of 650 kg/m$^3$, a thermal conductivity coefficient of 0.155 W/(m·°C), and a vapor permeability coefficient of 0.15 mg/(m·h·Pa). The inner layer of the finishing coating had a thickness of 0.1 m, while the outer layer had a thickness of 0.2 m.

Figure 1 illustrates the structural design of the investigated wall.

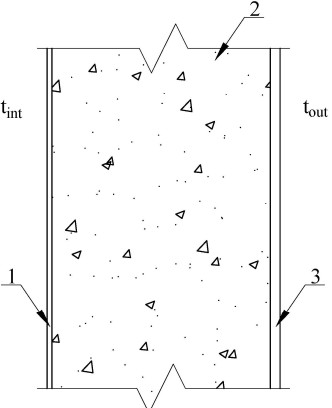

**Figure 1.** The calculation scheme for the wall is as follows: (1) internal finishing coating; (2) aerated concrete; (3) outer finishing coating.

The design parameters for the internal air were set at a temperature of 20.0 °C and a relative humidity of 55%. The design parameters for the outside air were determined according to the requirements in the "SP 131.13330.2020" SNiP 23-01-99 "Building climatology" (Table 1).

**Table 1.** Design climatic parameters.

| City | Rostov-on-Don | Voronezh | Novosibirsk | Vorkuta |
|---|---|---|---|---|
| Average temperature of the heating period $t_{hp}$, °C | −0.1 | −2.5 | −8.1 | −9.5 |
| Terms of use | A | A | A | A |
| Duration of the heating period $Z_{hp}$, days | 166 | 190 | 222 | 289 |
| Average monthly relative air humidity during the coldest month, % | 85 | 83 | 77 | 80 |
| Heating degree days | 3337 | 4275 | 6238 | 8791 |
| Monthly average temperatures used in the calculations: | | | | |
| October | 9.6 | 6.6 | 2.6 | −4.2 |
| November | 3.4 | 0 | −7.3 | −12.8 |
| December | −1.2 | −4.8 | −14.4 | −16.7 |
| January | −3.8 | −7.4 | −17.6 | −20.4 |
| February | −3.0 | −7.0 | −15.8 | −20.0 |
| March | 2.4 | −1.3 | −8.0 | −14.2 |
| April | 10.9 | 8.4 | 2.7 | −9.4 |

To determine the minimum allowable thickness of the aerated concrete layer in the building wall for each considered climatic condition, the elemental requirements for thermal protection in newly erected buildings were taken into account (Table 2).

**Table 2.** Thickness of the aerated concrete layer, m.

| City | Aerated Concrete Grade | | | |
|---|---|---|---|---|
| | D350 | D400 | D500 | D600 |
| Vorkuta | 0.60 | 0.65 | 0.80 | 0.95 |
| Novosibirsk | 0.45 | 0.50 | 0.65 | 0.75 |
| Voronezh | 0.40 | 0.40 | 0.50 | 0.60 |
| Rostov-on-Don | 0.35 | 0.35 | 0.45 | 0.55 |

The temperature at which condensation starts $t_{b.c}$, for the cases under consideration, was determined according to the specified procedure. The structure under consideration is conditionally divided into several vertical layers with a thickness of 0.01 m each. Then, the analytical method is used to determine how the temperature is distributed at the boundary of each layer in the thickness of the outer wall $\tau_{xi}$ °C, according to the formula:

$$\tau_{xi} = t_{int} - (t_{int} - t_{out}) \frac{R^t_{xi}}{R^t_0} \qquad (1)$$

where $t_{int}$ is the design's indoor air temperature, °C;

$t_{out}$ is the design's outdoor air temperature, °C;

$R^t_0$ is the total actual reduced thermal resistance in the outer wall, (m²·°C)/W; and

$R^t_{xi}$ is the total thermal resistance to heat transfer, starting from the internal air to a given section in the thickness of the outer wall, (m²·°C)/W.

After that, according to the calculated values for the temperatures $\tau_{xi}$, the values for the maximum elasticity of the water vapor $E_i$ are calculated using the formulas:

$$E_{xi} = 610.5 \exp\left(\frac{17.269 \cdot \tau_{xi}}{237.3 + \tau_{xi}}\right), \ \tau_{xi} \geq 0\ ^\circ\text{C} \tag{2}$$

$$E_{xi} = 610.5 \exp\left(\frac{21.875 \cdot \tau_{xi}}{265.5 + \tau_{xi}}\right), \ \tau_{xi} \leq 0\ ^\circ\text{C} \tag{3}$$

Using the thermodynamic analogy, the values for the actual elasticity of the water vapor at the boundaries of the individual layers $e_x$, Pa, should be determined by the formula:

$$e_{xi} = e_{int} - (e_{int} - e_{out})\frac{R^p_{xi}}{R^p_0} \tag{4}$$

where $e_{int}$ is the actual water vapor pressure of the indoor air, Pa;

$e_{out}$ is the actual water vapor pressure of the outdoor air, Pa;

$R^p_0$ is the total actual reduced vapor permeability resistance in the outer wall, m²·h·Pa/mg; and

$R^p_{xi}$ is the total vapor permeability resistance, starting from the internal air to a given section in the thickness of the outer wall, m²·h·Pa/mg.

Then, using the empirical method, the outside air temperature $t_{out}$ was chosen, at which the following condition was fulfilled in the enclosure in one of the layers:

$$e_{xi} \approx E_{xi} \tag{5}$$

This temperature is called the temperature at which condensation begins $t_{b.c.}$. When the outside temperature $t_{out}$ drops below this value, the conditions for the formation of condensate are created in the outer wall. When the outside temperature $t_{out}$ rises above this value, the conditions for the formation of condensate will not be created in the outer wall (Figure 2).

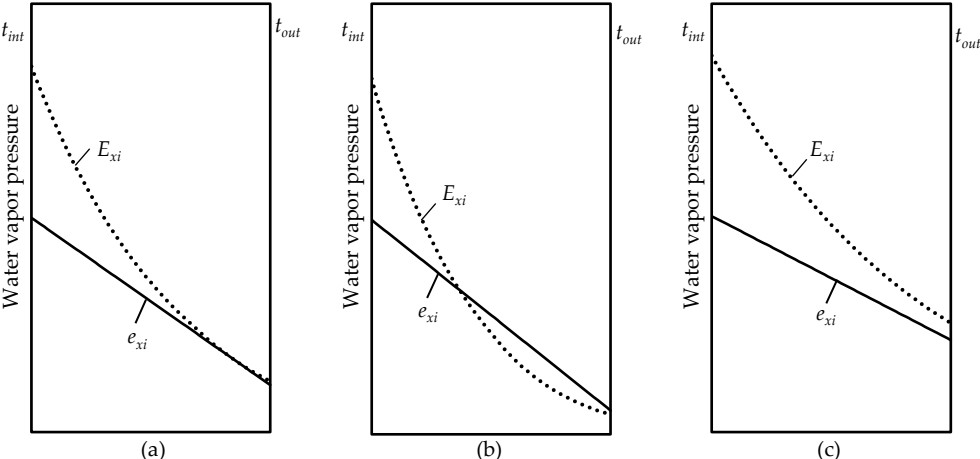

**Figure 2.** Method for determining the temperature at the onset of condensation $t_{b.c.}$: (**a**) $t_{out} = t_{b.c.}$; (**b**) $t_{out} < t_{b.c.}$; (**c**) $t_{out} > t_{b.c.}$.

To calculate the amount of condensate formed in a given month, the following method was employed. The amount of condensate falling per hour (mg/m²·h) was determined, using the formula, by the average temperature in the calculated month:

$$G^h_c = G^h_{hyd} - G^h_{dry} \tag{6}$$

where $G_{hyd}^h$ is the rate of diffusion of the water vapor during moistening, determined by the formula:

$$G_{hyd}^h = \frac{e_{int} - E_c}{R_{int}^p} \tag{7}$$

where $e_{in}$ is the actual elasticity of the water vapor in the internal air, Pa;

$E_c$ is the maximum elasticity of the water vapor in the plane of condensation, Pa; and

$R_{int}^p$ is the vapor permeability resistance of the layers located from the inner surface of the fence to the plane at the start of the condensation.

$G_{dry}^h$ is the rate of diffusion of the water vapor during drying, determined by the formula:

$$G_{dry}^h = \frac{E_c - e_{out}}{R_{out}^p} \tag{8}$$

where $e_{out}$ is the actual elasticity of the water vapor in the outside air, Pa; and

$R_p^{out}$ is the vapor permeability resistance of the layers from the plane, at the start of the condensation, to the outer surface of the enclosure, m²·h·Pa/mg.

Then, considering the duration of the specific month, the monthly amount of condensate falling was determined. Subsequently, the total amount of condensate falling for all months, with an average monthly temperature lower than the temperature at the condensation onset ($t_{b.c.}$), was determined.

The increase in the weight moisture ($W$ %), during the water vapor condensation, in the thickness of the aerated concrete, was calculated using the provided formula:

$$W = \frac{G_c}{\rho \cdot \delta} 100 \tag{9}$$

where $\rho$ is the volumetric mass of the wetted layer material, kg/m³; and

$\delta$ is the thickness of the condensation layer, m.

In total, the humidity regime for the 48 enclosed structures were analyzed throughout the course of this study. In this paper, we analyzed the obtained values for the temperature at the onset of the condensation $t_{b.c}$, the amount of condensate that fell out $G_c^h$, and the moisture content $W$ % in the aerated concrete, for all the considered wall structures. The obtained values were compared for various investigated enclosing structures. The influence of the density of the aerated concrete, the characteristics of the outer finishing coating, and the climatic conditions in the cities under consideration, on the humidity regime in the outer walls of the buildings were evaluated.

## 3. Results

Let us determine the temperature at the beginning of the condensation for all 48 considered external fences. To do this, we consistently use Formulas (1)–(5). In this article, we present the calculations summarized in Table 3. These calculations were performed for an external enclosing structure located in the city of Voronezh, using aerated concrete at the grade D400. Cement–sand plaster with a density of 1800 kg/m³ was used as an exterior finish in the structure.

**Table 3.** Thickness of the aerated concrete layer, m.

| Fencing Layers | Distance from the Inner Surface, cm | $R_{xi}^t$, m²·°C/W | $\tau_{xi}$, °C | $E_{xi}$, Pa | $R_{xi}^t$, m²·h·Pa/mg | $e_{xi}$, Pa | $E_{xi}$-$e_{xi}$, Pa |
|---|---|---|---|---|---|---|---|
| Inner surface of the wall | 0 | 0.115 | 19.13 | 2214 | 0.027 | 1273.8 | 940.6 |
| Lime–slag plaster | 1 | 0.136 | 18.97 | 2192 | 0.098 | 1243.0 | 949.4 |
| Aerated concrete D400 | 2 | 0.208 | 18.43 | 2120 | 0.142 | 1224.3 | 895.5 |
| | 3 | 0.279 | 17.90 | 2049 | 0.185 | 1205.5 | 843.8 |
| | 4 | 0.351 | 17.36 | 1981 | 0.228 | 1186.8 | 794.1 |

**Table 3.** *Cont.*

| Fencing Layers | Distance from the Inner Surface, cm | $R^t_{xi}$, m²·°C/W | $\tau_{xi}$, °C | $E_{xi}$, Pa | $R^t_{xi}$, m²·h·Pa/mg | $e_{xi}$, Pa | $E_{xi}$-$e_{xi}$, Pa |
|---|---|---|---|---|---|---|---|
| Aerated concrete D400 | 5 | 0.422 | 16.82 | 1914 | 0.272 | 1168.0 | 746.4 |
| | 6 | 0.493 | 16.28 | 1850 | 0.315 | 1149.3 | 700.7 |
| | 7 | 0.565 | 15.74 | 1787 | 0.359 | 1130.5 | 656.9 |
| | 8 | 0.636 | 15.20 | 1727 | 0.402 | 1111.8 | 614.9 |
| | 9 | 0.708 | 14.66 | 1668 | 0.446 | 1093.0 | 574.8 |
| | 10 | 0.779 | 14.12 | 1611 | 0.489 | 1074.3 | 536.5 |
| | 11 | 0.851 | 13.59 | 1555 | 0.533 | 1055.5 | 499.8 |
| | 12 | 0.922 | 13.05 | 1502 | 0.576 | 1036.8 | 464.9 |
| | 13 | 0.993 | 12.51 | 1450 | 0.620 | 1018.0 | 431.5 |
| | 14 | 1.065 | 11.97 | 1399 | 0.663 | 999.2 | 399.8 |
| | 15 | 1.136 | 11.43 | 1350 | 0.707 | 980.5 | 369.6 |
| | 16 | 1.208 | 10.89 | 1303 | 0.750 | 961.7 | 341.0 |
| | 17 | 1.279 | 10.35 | 1257 | 0.794 | 943.0 | 313.8 |
| | 18 | 1.351 | 9.82 | 1212 | 0.837 | 924.2 | 288.0 |
| | 19 | 1.422 | 9.28 | 1169 | 0.881 | 905.5 | 263.6 |
| | 20 | 1.493 | 8.74 | 1127 | 0.924 | 886.7 | 240.6 |
| | 21 | 1.565 | 8.20 | 1087 | 0.968 | 868.0 | 218.9 |
| | 22 | 1.636 | 7.66 | 1048 | 1.011 | 849.2 | 198.5 |
| | 23 | 1.708 | 7.12 | 1010 | 1.055 | 830.5 | 179.3 |
| | 24 | 1.779 | 6.58 | 973 | 1.098 | 811.7 | 161.4 |
| | 25 | 1.851 | 6.04 | 938 | 1.142 | 793.0 | 144.6 |
| | 26 | 1.922 | 5.51 | 903 | 1.185 | 774.2 | 129.0 |
| | 27 | 1.993 | 4.97 | 870 | 1.228 | 755.5 | 114.4 |
| | 28 | 2.065 | 4.43 | 838 | 1.272 | 736.7 | 101.0 |
| | 29 | 2.136 | 3.89 | 807 | 1.315 | 717.9 | 88.6 |
| | 30 | 2.208 | 3.35 | 776 | 1.359 | 699.2 | 77.3 |
| | 31 | 2.279 | 2.81 | 747 | 1.402 | 680.4 | 67.0 |
| | 32 | 2.351 | 2.27 | 719 | 1.446 | 661.7 | 57.6 |
| | 33 | 2.422 | 1.74 | 692 | 1.489 | 642.9 | 49.1 |
| | 34 | 2.493 | 1.20 | 666 | 1.533 | 624.2 | 41.6 |
| | 35 | 2.565 | 0.66 | 640 | 1.576 | 605.4 | 34.9 |
| | 36 | 2.636 | 0.12 | 616 | 1.620 | 586.7 | 29.2 |
| | 37 | 2.708 | −0.42 | 590 | 1.663 | 567.9 | 21.8 |
| | 38 | 2.779 | −0.96 | 564 | 1.707 | 549.2 | 14.9 |
| | 39 | 2.851 | −1.50 | 539 | 1.750 | 530.4 | 8.9 |
| | 40 | 2.922 | −2.04 | 516 | 1.794 | 511.7 | 3.9 |
| | 41 | 2.993 | −2.57 | 493 | 1.837 | 492.9 | −0.1 |
| Cement–sand plaster | 42 | 3.007 | −2.67 | 489 | 1.948 | 445.0 | 43.8 |
| | 43 | 3.020 | −2.77 | 485 | 2.059 | 397.1 | 87.6 |
| Outside air | | 3.063 | −3.10 | 471 | 2.073 | 391.3 | 80.1 |

The distribution of the water vapor pressure, at an outdoor temperature of $t_{out}$ = −3.1, equal to the temperature at the beginning of the condensation $t_{b.c.}$, for this outer wall, is presented in Figure 3.

The findings from the investigations, conducted to identify the temperature at which condensation begins ($t_{b.c.}$) for each of the examined 48 enclosed structures, are displayed in Table 4.

It has been observed that as the thickness of aerated concrete increases, the $t_{b.c.}$ decreases, except for the enclosing structure in the city of Vorkuta using the developed DBM as an exterior finish. Additionally, it has been found that higher densities of aerated concrete and outer finishing coatings result in lower $t_{b.c.}$.

The amount of condensed moisture that accumulates in the fence during the winter period was determined by comparing the $t_{b.c.}$ with the average temperature for each month from October to April. If the $t_{b.c.}$ for a specific fence variant exceeds the average monthly temperature for any given month, condensate will form during that month. Let us compare the temperature at the onset of condensation, $t_{b.c.}$, for the outer walls of a building located in the city of Voronezh with the average monthly temperatures (Figure 4).

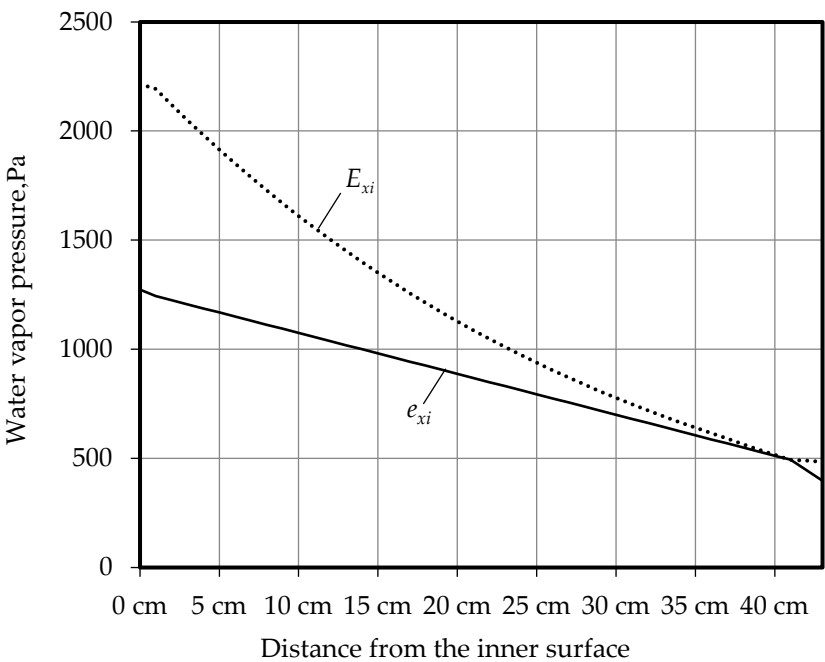

**Figure 3.** Distribution of the water vapor pressure along the wall thickness.

**Table 4.** Temperature at the beginning of the condensation $t_{b.c.}$, °C.

| City | Finishing Coating Density ρ, kg/m³ | Aerated Concrete Grade | | | |
|---|---|---|---|---|---|
| | | **D350** | **D400** | **D500** | **D600** |
| Rostov-on-Don | 650 | −7.9 | −8.7 | −10.4 | −11.3 |
| | 1100 | −2.5 | −3.4 | −6.6 | −8.4 |
| | 1800 | −0.9 | −1.7 | −4.7 | −7.1 |
| Voronezh | 650 | −8.7 | −9.3 | −10.6 | −11.5 |
| | 1100 | −4.0 | −4.8 | −7.5 | −8.9 |
| | 1800 | −2.4 | −3.1 | −6.0 | −7.8 |
| Novosibirsk | 650 | −9.9 | −10.5 | −11.4 | −12.0 |
| | 1100 | −6.6 | −7.6 | −9.2 | −10.1 |
| | 1800 | −5.2 | −6.5 | −8.4 | −9.4 |
| Vorkuta | 650 | −9.8 | −10.2 | −11 | −11.5 |
| | 1100 | −7.1 | −7.9 | −9.2 | −10 |
| | 1800 | −6.0 | −8.9 | −8.4 | −9.3 |

It has been established that conditions for the formation of condensate will not be created in walls made of D600 grade aerated concrete blocks. In the walls made of aerated concrete blocks of grades D350 and D400, plastered with a cement–sand composition or Knauf GRUNBAND, condensation will occur at the average monthly temperatures for December, January, and February (Figure 4, curves 1,2). In the walls of aerated concrete blocks made of the D500 grade, plastered with a cement–sand composition, condensation will occur at the average monthly temperatures for January and February. In the walls of aerated concrete blocks made of the D500 grade, plastered with Knauf GRUNBAND, conditions for the formation of condensate will not be created. Thus, for the conditions in the city of Voronezh, the use of heat-insulating lime plaster can significantly improve the humidity regime in walls made of aerated concrete blocks of grades D350, D400, and D500. Similarly, the same comparisons were made for the cities of Rostov-on-Don, Novosibirsk, and Vorkuta.

This calculation was performed for an external enclosing structure located in the city of Voronezh, with aerated concrete at grade D400. Cement–sand plaster with a density of

1800 kg/m³ was used as an exterior finish in the structure. To do this, we consistently use Formulas (6)–(8). The calculation results are presented in Table 5.

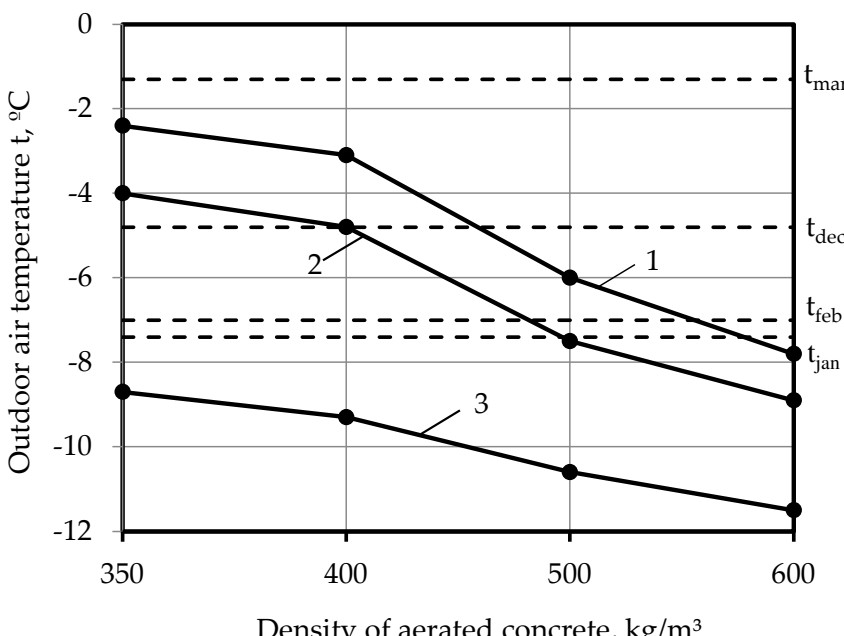

**Figure 4.** Dependence of the temperature at the beginning of the condensation, $t_{b.c.}$, on the density of the aerated concrete in the city of Voronezh: (1) cement–sand plaster; (2) Knauf GRUNBAND; (3) the developed DBM.

**Table 5.** Determination on the amount of condensing moisture per year.

| Name of the Month | January | February | March | April | October | November | December |
|---|---|---|---|---|---|---|---|
| Month number | I | II | III | IV | X | XI | XII |
| Average temperature of the month $t_{out}$, °C | −7.4 | −7.0 | −1.3 | 8.4 | 6.6 | 0.0 | −4.8 |
| Diffusion rate during humidification $G_{hyd}^h$, mg/m²·h | 512 | 506 | 0.0 | 0.0 | 0.0 | 0.0 | 467 |
| Drying diffusion rate $G_{dry}^h$, mg/m²·h | 313 | 322 | 0.0 | 0.0 | 0.0 | 0.0 | 380 |
| Amount of condensate falling out per hour $G_c^h$, mg/m² | 199 | 184 | 0.0 | 0.0 | 0.0 | 0.0 | 87 |
| Amount of moisture (moisture) per month $G_{hyd}^{mon}$, kg/m² | 0.381 | 0.377 | 0.0 | 0.0 | 0.0 | 0,0 | 0.347 |
| Amount of moisture (drying) per month $G_{dry}^{mon}$, kg/m² | 0.233 | 0.239 | 0.0 | 0.0 | 0.0 | 0.0 | 0.283 |
| Amount of falling condensate per month $G_c^{mon}$, kg/m² | 0.148 | 0.138 | 0.0 | 0.0 | 0.0 | 0.0 | 0.064 |
| Amount of condensate falling out for the entire period of moisture accumulation per year, kg/m² | | | | | | | 0.350 |

Similarly, the calculation was carried out for the remaining 48 studied external enclosing structures. The research results for the cities of Rostov-on-Don and Voronezh are shown in Figure 5.

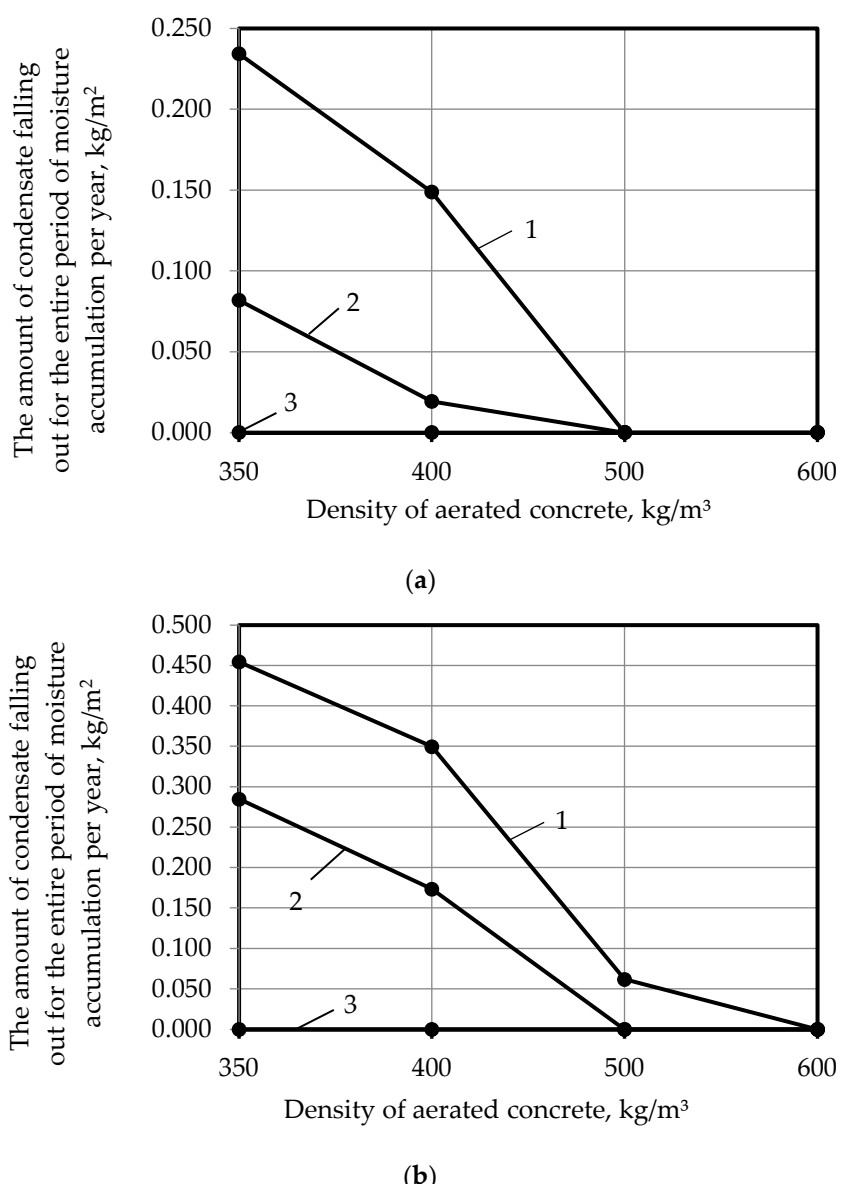

**Figure 5.** Dependence of the amount of condensate falling out for the entire period of moisture accumulation per year, kg/m$^2$, on the density of the aerated concrete: (**a**) for the city of Rostov-on-Don; (**b**) for the city of Voronezh; (1) cement–sand plaster; (2) Knauf GRUNBAND; (3) the developed DBM.

For the city of Rostov-on-Don (Figure 5a), it has been established that condensate will form when finishing aerated concrete blocks of grades D350 and D400 with cement–sand plaster and a DBM of Knauf GRUNBAND. However, the use of the developed DBM can prevent condensate formation regardless of the aerated concrete grade. Condensate will only form during the winter months in Rostov-on-Don.

In the conditions in Voronezh (Figure 5b), condensate will form when finishing aerated concrete blocks of grades D350, D400, and D500 with cement–sand plaster, as well as when finishing aerated concrete blocks of grades D350 and D400 with a DBM of Knauf GRUNBAND. The amount of condensate formed during the finishing of aerated concrete blocks D350 with a DBM of Knauf GRUNBAND is 3.5 times higher in Voronezh compared to Rostov-on-Don. Similar to Rostov-on-Don, the use of the developed DBM can prevent condensate formation regardless of the aerated concrete grade. Condensate will also only form during the winter months in Voronezh.

The research results for the cities of Novosibirsk and Vorkuta are presented in Figure 6.

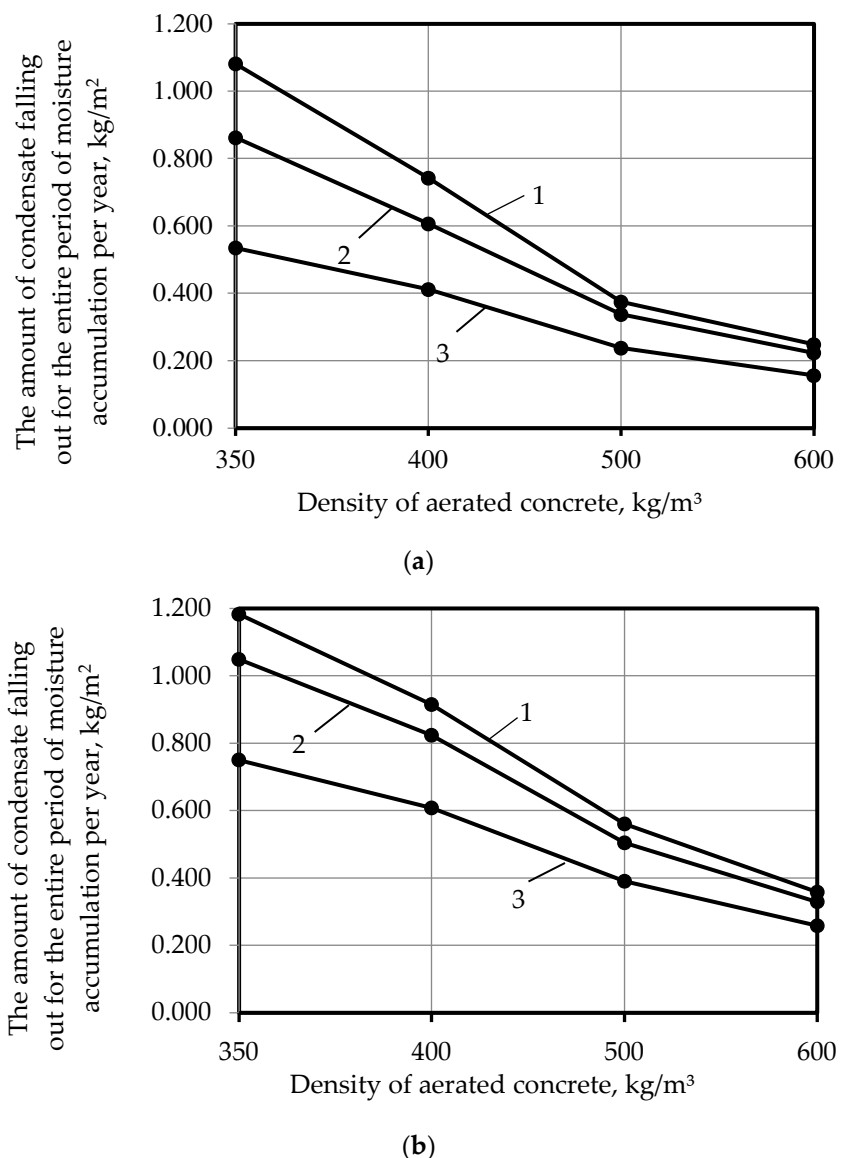

**Figure 6.** Dependence of the amount of condensate falling out for the entire period of moisture accumulation per year, kg/m², on the density of the aerated concrete: (**a**) for the city of Novosibirsk; (**b**) for the city of Vorkuta; (1) cement–sand plaster; (2) Knauf GRUNBAND; (3) the developed DBM.

In Novosibirsk (Figure 6a), condensate will form in all the structures being considered. However, the use of the developed DBM can significantly reduce the amount of condensate. Compared to Knauf GRUNBAND plaster, the use of the DBM reduces condensate formation by 38.0% for D350 aerated concrete and by 50.6% for cement–sand plaster. For D400 aerated concrete, the reduction is 32.0% compared to Knauf GRUNBAND plaster and 45.5% compared to cement–sand plaster. For D500 aerated concrete, the reduction is 29.6% compared to Knauf GRUNBAND plaster and 36.5% compared to cement–sand plaster. For D600 aerated concrete, the reduction is 30.0% compared to Knauf GRUNBAND plaster and 37.1% compared to cement–sand plaster. It is important to note that condensate will occur not only in the winter months, but also in March and November.

In Vorkuta (Figure 6b), condensate will also form in all the structures being considered. However, the use of the developed DBM can reduce the amount of condensate compared to Knauf GRUNBAND plaster and cement–sand plaster. The reduction ranges from 21.5% to 28.5% depending on the grade of aerated concrete when compared to Knauf GRUN-BAND plaster, and from 27.9% to 36.6% depending on the grade of aerated concrete when

compared to cement–sand plaster. Similar to Novosibirsk, condensate will not only occur in the winter months, but also in March, April, and November. It should be noted that the thickness of the condensation layer varies for different fencing options and outdoor temperatures for both cities.

To be able to compare different structures, the thickness of the aerated concrete layer was taken for two options for the calculation: for the entire thickness of the aerated concrete and for a layer of the aerated concrete with a thickness of 0.05 m at the boundary of the outer finishing coating/aerated concrete (the largest amount of condensate will accumulate in the aerated concrete layer at the border with the plaster coating). To do this, we use Formula (9).

The results from the studies, carried out to determine the weight moisture content of the aerated concrete % for each of the structures in the study, are presented in Table 6.

**Table 6.** Increase in weight moisture, %.

| City | Finishing Coating Density ρ, kg/m³ | Aerated Concrete Grade | | | |
|---|---|---|---|---|---|
| | | D350 | D400 | D500 | D600 |
| Rostov-on-Don | 650 | 0.00 * | 0.00 | 0.00 | 0.00 |
| | | 0.00 | 0.00 | 0.00 | 0.00 |
| | 1100 | 0.47 | 0.10 | 0.00 | 0.00 |
| | | 0.07 | 0.014 | 0.00 | 0.00 |
| | 1800 | 1.34 | 0.74 | 0.00 | 0.00 |
| | | 0.19 | 0.11 | 0.00 | 0.00 |
| Voronezh | 650 | 0.00 | 0.00 | 0.00 | 0.00 |
| | | 0.00 | 0.00 | 0.00 | 0.00 |
| | 1100 | 1.63 | 0.87 | 0.00 | 0.00 |
| | | 0.20 | 0.11 | 0.00 | 0.00 |
| | 1800 | 2.60 | 1.75 | 0.25 | 0.00 |
| | | 0.32 | 0.22 | 0.02 | 0.00 |
| Novosibirsk | 650 | 3.06 | 2.06 | 0.95 | 0.52 |
| | | 0.34 | 0.21 | 0.07 | 0.035 |
| | 1100 | 4.93 | 3.03 | 1.35 | 0.74 |
| | | 0.55 | 0.30 | 0.10 | 0.05 |
| | 1800 | 6.18 | 3.71 | 1.50 | 0.83 |
| | | 0.69 | 0.37 | 0.12 | 0.055 |
| Vorkuta | 650 | 4.29 | 3.04 | 1.56 | 0.86 |
| | | 0.36 | 0.23 | 0.10 | 0.045 |
| | 1100 | 5.99 | 4.12 | 2.02 | 1.10 |
| | | 0.50 | 0.32 | 0.13 | 0.058 |
| | 1800 | 6.76 | 4.58 | 2.24 | 1.19 |
| | | 0.56 | 0.35 | 0.14 | 0.063 |

Note *: above the line relates to an increase in weight moisture W,% for a layer of aerated concrete with a thickness of 0.05 m at the boundary of the outer finishing coating/aerated concrete; under the line relates to an increase in weight moisture W,% for the entire thickness of the aerated concrete.

The utilization of the developed DBM results in a decrease in the condensate volume and a reduction in the moisture weight increase percentage in aerated concrete fences of grades D350-D600.

## 4. Conclusions

In total, during the study, the humidity regime for 48 wall enclosing structures was analyzed. It was found that the use of the developed DBM makes it possible to reduce the temperature at the onset of condensation $t_{b.c.}$ by 1.5–7.0 °C and reduce the amount of condensate $G_c^h$ by 21.5–50.6%. Thus, the use of aerated concrete developed using a DBM for exterior finishing will allow for the maintenance of high thermal insulation properties of the exterior walls throughout the entire service life by improving the humidity regime Inside the wall. This will significantly reduce energy consumption for heating during the operation of aerated concrete buildings throughout the entire life cycle of such buildings. Reducing

the consumption of energy resources will reduce the negative impact on the environment from the extraction, transportation, and combustion of energy resources. In addition, the use of the developed DBM will extend the service life of the exterior finishing materials and increase the duration of the non-repair operation of aerated concrete buildings. At the same time, it is important to note that the developed DBM is especially important to use when finishing aerated concrete with a density relating to D350-D400, since when using aerated concrete of these grades, the increase in weight moisture is more than two times higher compared to aerated concrete grades D500-D600. Also, when using the developed DBM, aerated concrete production waste will be efficiently processed. The results obtained are in good agreement with the results from other studies on similar topics [28–30].

The novelty of the research lies in the complex of findings obtained, reflecting the influence of the characteristics of the outer finishing coating, the density of aerated concrete, and the climatic characteristics of the humidity regime inside the outer walls. Based on the scientific novelty of our study, the following shortcomings in the current regulatory documentation governing the choice of exterior finishing coatings for aerated concrete can be distinguished:

- There are no requirements for the thermal conductivity of external finishing coatings, while this characteristic significantly affects the humidity regime in external fences;
- Normalized values on average density and resistance to vapor permeability are given without taking into account the brand of aerated concrete. At the same time, the humidity regime in the walls made of aerated concrete of various grades will differ significantly;
- Normalized values on average density and resistance to vapor permeability are given without taking into account the climatic characteristics of the construction area. At the same time, the humidity regime in the walls made of aerated concrete will very much depend on the construction area.

Taking into account the results from our study and the identified shortcomings in the current regulatory documentation, we propose the following areas for further research:

- It is necessary to develop mathematical dependences of the temperature at the onset of condensation and the amount of condensate falling out in the aerated concrete enclosing structure on the thermal conductivity and vapor permeability of the outer finishing coating;
- To develop a methodology that allows the assessment, with high accuracy, of the humidity regime in aerated concrete walls, depending on the characteristics of the external finishing coating and the climatic conditions, which will make it possible to provide recommendations for finalizing the current regulatory documentation;
- In order to assess the actual effects on the environment, it is necessary to carry out studies to determine the experimental values for the moisture content of aerated concrete during the life cycle of buildings constructed from aerated concrete and finished with the developed DBM;
- It is necessary to evaluate the savings in energy resources spent on heating buildings by improving the humidity regime in the outer walls throughout the entire life cycle of the building.

We believe that the proposed method for determining the amount of condensate falling out per year, taking into account the temperature at the onset of condensation $t_{b.c.}$, and the temperature for each of the months in terms of moisture accumulation, can also be used for other wall structures.

**Author Contributions:** Conceptualization, M.V.F.; Methodology, M.V.F.; Data curation, E.A.Z.; Writing—review & editing, V.I.L. All authors have read and agreed to the published version of the manuscript.

**Funding:** This research received no external funding.

**Data Availability Statement:** The data that support the findings of this study are available on request from the corresponding author.

**Conflicts of Interest:** The authors declare no conflict of interest.

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
