# Peer review of "Humidity Conditions of Aerated Concrete Walls Depending on the Type of Finishing Coating"

_applsci, doi:10.3390/app13179529_

Round 1
Reviewer 1 Report

minor
Author Response
Response to Reviewer 1 Comments
Point 1: I think the Abstract needs to be extended a little and to be included some statistical data of the dry building mixture as the best finding.
Response 1: The abstract has been expanded, the experimental values obtained during the work have been added.
Point 2: In addition, ‘dry building mix’ can be amended to ‘dry building mixture’ in the Abstract.
Response 2: Necessary changes have been made in the Abstract.
Point 3: Fig 1 looks familiar to me and it seems it is copied from a similar article. That’s why the quality is lower compared to the other Figures. Please re-draw it.
Response 3: Figure 1 redrawn, quality improved.
Point 4: What are the aerated concrete grades based on? Any standards?
Response 4: The answer to the question has been added to the introduction of the article. «The work investigated the humidity regime in the walls of aerated concrete D350-600. This choice of aerated concrete density is due to the fact that aerated concrete of these grades is usually used in the construction of single-layer walls of aerated concrete.»
Point 5: I suggest changing ‘figure’ to ‘Figure’ in the context.
Response 5: " figure " changed to " Figure " in context.
Point 6: Try to cite more up-to-date articles in regard to aerated concrete.
Response 6: References corrected

Reviewer 2 Report
In this study, a study was conducted to analyze how different types of plaster coatings affect the hu-10 midity levels in aerated concrete walls under varying climatic conditions. All of the following comments should be addressed in detail and responded to in an appropriate manner so I can recommend the paper for publication:
(1) The authors should explain or annotate the abbreviations at their first use site. It is beneficial to readers who are not familiar with this field. E.g. line 59, Radg=0.71 MPa, vapor permeability coefficient μ=0.15 mg/(m∙h∙Pa ),
(2) Please indicate the standard deviation and relevant statistical tests to validate the conclusions.
(3) Methods section determines the results. Kindly focus on three basic elements of the methods section. a. How the study was designed? b. How the study was carried out? c. How the data were analyzed?
(4) There have been many studies investigating the effect of humidity on the performance of air-entrained concrete, and the authors should discuss them in the Introduction. E.g. A green ultra-lightweight chemically foamed concrete for building exterior: A feasibility study; The roles of cenosphere in ultra-lightweight foamed geopolymer concrete (UFGC)
(5) Please make sure your conclusions' section underscore the scientific value added of your paper, and/or the applicability of your findings/results, as indicated previously. Please revise your conclusion part into more details. Basically, you should enhance your contributions, limitations, underscore the scientific value added of your paper, and/or the applicability of your findings/results and future study in this session.
(6) Please emphasize the novelty of this study in the Introduction.
In this study, a study was conducted to analyze how different types of plaster coatings affect the hu-10 midity levels in aerated concrete walls under varying climatic conditions. All of the following comments should be addressed in detail and responded to in an appropriate manner so I can recommend the paper for publication:
(1) The authors should explain or annotate the abbreviations at their first use site. It is beneficial to readers who are not familiar with this field. E.g. line 59, Radg=0.71 MPa, vapor permeability coefficient μ=0.15 mg/(m∙h∙Pa ),
(2) Please indicate the standard deviation and relevant statistical tests to validate the conclusions.
(3) Methods section determines the results. Kindly focus on three basic elements of the methods section. a. How the study was designed? b. How the study was carried out? c. How the data were analyzed?
(4) There have been many studies investigating the effect of humidity on the performance of air-entrained concrete, and the authors should discuss them in the Introduction. E.g. A green ultra-lightweight chemically foamed concrete for building exterior: A feasibility study; The roles of cenosphere in ultra-lightweight foamed geopolymer concrete (UFGC)
(5) Please make sure your conclusions' section underscore the scientific value added of your paper, and/or the applicability of your findings/results, as indicated previously. Please revise your conclusion part into more details. Basically, you should enhance your contributions, limitations, underscore the scientific value added of your paper, and/or the applicability of your findings/results and future study in this session.
(6) Please emphasize the novelty of this study in the Introduction.
Author Response
Response to Reviewer 2 Comments
Point 1: The authors should explain or annotate the abbreviations at their first use site. It is beneficial to readers who are not familiar with this field. E.g. line 59, Radg=0.71 MPa, vapor permeability coefficient μ=0.15 mg/(m∙h∙Pa ).
Response 1: Abbreviations are commented
Point 2: Please indicate the standard deviation and relevant statistical tests to validate the conclusions.
Response 2: Deviations are indicated in the introduction. The studies were carried out using a numerical experiment, so we believe that there is no need to indicate the standard deviation and the corresponding statistical tests.
Point 3: Methods section determines the results. Kindly focus on three basic elements of the methods section. a. How the study was designed? b. How the study was carried out? c. How the data were analyzed?
Response 3: The methods section has been supplemented in accordance with the recommendations.
Point 4: There have been many studies investigating the effect of humidity on the performance of air-entrained concrete, and the authors should discuss them in the Introduction. E.g. A green ultra-lightweight chemically foamed concrete for building exterior: A feasibility study; The roles of cenosphere in ultra-lightweight foamed geopolymer concrete (UFGC)
Response 4: The introduction section has been revised taking into account the comments.
Point 5: Please make sure your conclusions' section underscore the scientific value added of your paper, and/or the applicability of your findings/results, as indicated previously. Please revise your conclusion part into more details. Basically, you should enhance your contributions, limitations, underscore the scientific value added of your paper, and/or the applicability of your findings/results and future study in this session.
Response 5: The conclusions section has been revised taking into account the comments.
Point 6: Please emphasize the novelty of this study in the Introduction.
Response 6: The introduction section has been revised taking into account the comments.

Reviewer 3 Report
The number of self-citations should be reduced.The work must be submitted for review by a native English speaker.
Author Response
Response to Reviewer 3 Comments
Point 1: The number of self-citations should be reduced.
Response 1: References corrected. The number of references has been increased from 25 to 31. The number of self-citations has been reduced from 6 to 5.
